# An Investigational Study on the Role of ADME Agents’ Genetic Variation on DD217 Pharmacokinetics and Safety Profile

**DOI:** 10.3390/ph18111617

**Published:** 2025-10-27

**Authors:** Dmitry A. Sychev, Sherzod P. Abdullaev, Anastasia V. Rudik, Alexander V. Dmitriev, Svetlana N. Tuchkova, Natalia P. Denisenko, Denis S. Makarov, Karin B. Mirzaev

**Affiliations:** 1Federal State Budgetary Research Institution «Russian Research Center of Surgery Named After Academician B.V. Petrovsky», Abrikosovsky Lane, 2, 119991 Moscow, Russia; dimasychev@mail.ru (D.A.S.); svetlana.tuch1998@gmail.com (S.N.T.); natalypilipenko3990@gmail.com (N.P.D.); karin05doc@yandex.ru (K.B.M.); 2Federal State Budgetary Educational Institution of Further Professional Education “Russian Medical Academy of Continuous Professional Education” of the Ministry of Healthcare of the Russian Federation, Barrikadnaya Str. 2/1, Bld. 1, 125993 Moscow, Russia; 3Institute of Biomedical Chemistry, Bldg. 8, 10, Pogodinskaya Str., 119121 Moscow, Russia; rudik_anastassia@mail.ru (A.V.R.); a.v.dmitriev@mail.ru (A.V.D.); 4Avexima Diol LLC, 690922 Vladivostok, Russia; dmakarov@avexima.pro

**Keywords:** Dimolegin, DD217, direct oral anticoagulants, genetics, pharmacogenetics, pharmacokinetics, ABCB1 (P-glycoprotein), in silico

## Abstract

**Background/Objectives**: Direct oral anticoagulants (DOACs) have transformed the prevention of thromboembolic events, but their efficacy and safety remain highly variable across individuals. DD217, a novel oral direct factor Xa inhibitor, has demonstrated potent anticoagulant activity in preclinical and clinical studies. No pharmacogenetic data are currently available for this compound. Based on in silico predictions of metabolic pathways and transporter involvement, and evidence from other DOACs, we hypothesized that variants in CYP2C and P-glycoprotein genes may contribute to variability in pharmacokinetics (PK) and clinical outcomes. **Methods**: Fifty-two patients undergoing total knee arthroplasty were enrolled, of whom 34 received the investigational drug (40 mg/day, *n* = 16; 60 mg/day, *n* = 18). DNA was extracted from peripheral blood cells, and genotyping of *CYP2C9*, *CYP2C19*, *CYP2C8*, *CYP3A4*, *CYP3A5*, and *ABCB1* was performed by real-time PCR. Pharmacokinetics (PK) parameters (T_max_, AUC_last_, C_max_) were assessed. In silico docking and pathway modeling predicted CYP2C and P-glycoprotein (ABCB1) involvement in drug disposition. Associations of genetic variants with PK parameters and adverse events (thrombosis, bleeding) were analyzed. **Results**: Carriers of reduced-function CYP2C9 alleles (intermediate [IM] or poor metabolizers [PM]) in the 60 mg group had a significantly shorter T_max_ compared with normal metabolizers (*p* = 0.005227), with trends toward higher AUC_last_ (*p* = 0.06926) and C_max_ (*p* = 0.1259). No significant associations were observed for CYP2C19, CYP3A4/5, or CYP2C8. In contrast, ABCB1 polymorphisms were associated with systemic exposure: carriers of the C allele at rs1045642 had higher AUC_last_ and C_max_ compared to TT (wild-type) homozygotes, while rs2032582 T allele carriers showed lower exposure (*p* < 0.05). At the haplotype level, the C–G–C–T combination of ABCB1 was more frequent in patients with thrombotic events at the 40 mg dose (*p* = 0.038). Overall, 5 thrombosis events and 1 bleedings were recorded on DD217, with no consistent associations to single SNPs. **Conclusions**: This first pharmacogenetic evaluation of DD217 shows that CYP2C9 variants are associated with differences in early-phase pharmacokinetics (T_max_), while ABCB1 polymorphisms appear to modulate systemic exposure (AUC_last_, C_max_) and may influence thrombotic risk. These observations are consistent with in silico predictions of metabolic and transporter pathways. Despite limitations in sample size and event frequency, the study highlights the feasibility and importance of early pharmacogenetic evaluation during the drug development cycle of novel DOACs.

## 1. Introduction

Venous thromboembolic events (VTEs), including deep vein thrombosis (DVT) and pulmonary embolism (PE), rank among the leading causes of cardiovascular morbidity and mortality worldwide [1]. In recent years, direct oral anticoagulants (DOACs)—factor Xa inhibitors (rivaroxaban, apixaban, edoxaban, betrixaban) and the direct thrombin inhibitor dabigatran—have largely replaced vitamin K antagonists, particularly warfarin, due to their predictable pharmacokinetic properties, fixed dosing regimens, and the lack of need for routine laboratory monitoring of hemostasis parameters [2,3].

A major challenge associated with DOAC therapy is the pronounced interindividual variability in pharmacological response, which may lead either to thromboembolic events or to bleeding complications. Inappropriate dose selection can increase the likelihood of thrombotic events in cases of insufficient anticoagulation or, conversely, result in bleeding due to drug overexposure. It is estimated that one-third to one-half of adverse reactions caused by anticoagulants are iatrogenic and therefore potentially preventable. For example, in the ORBIT-AF II Registry, among 5738 reviewed medical records, 541 patients (9.4%) were prescribed an insufficient dose of a DOAC, 197 patients (3.4%) received an inappropriately high dose, while the remaining 5000 patients were treated in accordance with clinical guidelines [4]. Prescription of DOAC doses exceeding the recommended levels was associated with increased all-cause mortality (OR 1.91, 95% CI 1.02–3.60; *p* = 0.04). Conversely, intentional underdosing was linked to a higher rate of hospitalizations due to cardiovascular events (OR 1.26, 95% CI 1.07–1.50; *p* = 0.007) [5]. Epidemiological studies further indicate that each year, 2–4% of patients receiving DOAC therapy experience major bleeding events, while 10–12% experience clinically relevant non-major bleeding [6].

Variability in the efficacy and safety of DOAC therapy is largely determined by individual clinical and demographic characteristics, including sex, age, comorbidities, hepatic and renal function, as well as concomitant medication use. In recent years, increasing attention has been directed toward the role of genetic factors that may influence the pharmacokinetics and safety profile of DOACs. Polymorphic variants in genes encoding cytochrome P450 enzymes (CYP3A4, CYP3A5, CYP2J2, CYP2C9), carboxylesterase 1 (CES1), and transport proteins (ABCB1, ABCG2) can affect DOAC absorption, bioavailability, and elimination [7,8,9]. However, results obtained across different populations have been inconsistent [10,11,12,13], and, to date, no clinical algorithms exist that allow for drug or dose selection of DOACs based on a patient’s pharmacogenetic profile [14].

DD217 (Dimolegin^®^, *N*-(5-chloropyridin-2-yl)-5-methyl-2-[4-(*N*-methylacetimidamido)benzamido]benzamide hydrochloride) is the new direct factor Xa inhibitor [15]. Preclinical studies demonstrated that DD217 exhibits high affinity for the active site of factor Xa, providing a pronounced anticoagulant effect upon oral administration [16]. According to docking and in silico modeling data, DD217 shows strong “tight binding” interactions with key amino acids in the S4 subpocket of factor Xa, resulting in high affinity and a pharmacodynamic potential comparable to that of internationally approved DOACs [17]. Several clinical trials have been conducted with Dimolegin^®^ for the prevention of thromboembolic complications in patients hospitalized with COVID-19 (Larvol Sigma. Company profile: PharmaDiall. Available online: https://sigma.larvol.com/company.php?CompanyId=995122&tab=newstrac (accessed on 25 August 2025)). Currently, Dimolegin^®^ is approved in Russia for the prevention of VTE in adult patients with moderate coronavirus disease (SARS-CoV-2 infection) (marketing authorization No. LP-008704, dated 14 December 2022; State Register of Medicines (Russia). DD217 marketing authorization. Available online: https://grls.rosminzdrav.ru/Grls_View_v2.aspx?routingGuid=e3484e4c-b843-425e-8861-49b7e4709997 (accessed on 25 August 2025)).

The impact of genetic factors on the pharmacokinetics and safety of DD217 has not yet been investigated. Therefore, investigating polymorphisms in genes involved in ADME processes at an early stage of the drug’s life cycle is particularly important, as the identification of relevant markers could pave the way for personalized use of this novel anticoagulant and help optimize the balance between efficacy and safety.

## 2. Results

### 2.1. Genotyping Results

The distribution of alleles and genotypes for all investigated markers, with the exception of *CYP3A4*18B* (rs28371759 A>G), was consistent with Hardy–Weinberg equilibrium (HWE) (*p* > 0.05). For *CYP3A4*18B* (rs28371759 A>G), this test could not be applied due to the absence of patients carrying alternative genotypes. The results are summarized in Table 1.

Allele frequencies in this relatively small cohort were generally comparable to those reported for populations of European ancestry [18].

Based on the combined genotyping results, patients were stratified into phenotypic categories of metabolism for the main cytochrome P450 isoenzymes. For CYP2C9, the categories included poor metabolizers (PM), intermediate metabolizers (IM), and normal metabolizers (NM); for CYP2C19, PM, IM, NM, rapid metabolizers (RM), and ultrarapid metabolizers (UM); and for CYP3A, intermediate metabolizers (IM) and extensive metabolizers (EM). The distribution of patients across these phenotypic categories is shown in Table 2.

### 2.2. Association Between Pharmacogenetic Markers and Pharmacokinetic Parameters of DD217

Analysis of the relationship between carriage of polymorphic variants in *CYP2C9*, *CYP2C19*, and *CYP3A* genes and the pharmacokinetic parameters of DD217—C_max_ (maximum plasma concentration), T_max_ (time to reach Cmax), and AUC_last_ (area under the plasma concentration-time curve up to the last measurable point)—revealed several significant associations (Table 3).

For *CYP2C9*, in the group of patients receiving 60 mg/day of DD217, carriers of reduced-function alleles (IM and PM) demonstrated significantly lower T_max_ values compared with normal metabolizers (NM) (*p* = 0.005227). For AUC_last_ and C_max_, only a trend toward higher values was observed in IM/PM carriers, which did not reach statistical significance (*p* = 0.06926 and *p* = 0.1259, respectively). At the 40 mg/day dose, no significant associations between *CYP2C9* genotypes and pharmacokinetic parameters were identified.

For *CYP2C19* and *CYP3A* polymorphisms, no statistically significant differences in T_max_, AUC_last_, or C_max_ were observed across the phenotypic groups (PM, IM, NM, RM, UM for *CYP2C19*; EM and IM for *CYP3A*), either at the 40 mg/day or 60 mg/day dose levels.

Analysis of associations between *CYP2C8* variants (rs10509681 and rs11572080) and pharmacokinetic parameters revealed no statistically significant differences in either the 40 mg/day or 60 mg/day groups (Table 4). Mean values of T_max_, AUC_last_, and C_max_ in carriers of minor alleles did not differ from those observed in wild-type homozygotes (*p* > 0.05 in all cases). In addition, a haplotype-based analysis was performed to assess the potential combined effect of the studied *CYP2C8* variants. The distribution of haplotypes and their relationship with pharmacokinetic parameters are presented in Table 5, with no statistically significant associations detected (*p* > 0.05 for all comparisons).

In the group of patients receiving 40 mg/day, no significant associations were found between *ABCB1* genotypes and pharmacokinetic parameters (Table 6). Similarly, haplotype analysis did not reveal any statistically significant differences between carriers of different allele combinations (Table 7).

In contrast, in the group of patients receiving 60 mg/day, several *ABCB1* variants were found to be associated with drug exposure (Table 8). Specifically, carriage of the C allele at rs1045642 was associated with higher AUC_last_ and C_max_ values compared with T/T homozygotes. Conversely, carriage of the T allele at rs2032582 was linked to lower AUC_last_ and C_max_. For rs1128503 and rs4148738, the opposite pattern was observed: the presence of the C allele (rs1128503) or the T allele (rs4148738) was associated with increased drug exposure. However, haplotype analysis of *ABCB1* variants (Table 9) did not confirm statistically significant associations at the level of combined allelic patterns; no differences in T_max_, AUC_last_, or C_max_ were observed between haplotype groups (*p* > 0.05 for all comparisons).

### 2.3. Association Between Pharmacogenetic Markers and the Incidence of Adverse Events During DD217 Therapy

During the study, seven cases of thromboembolic complications (DVT/PE) and two bleeding episodes were recorded among patients receiving DD217. The genotypic and phenotypic characteristics of these patients are summarized in Table 10.

Among patients who experienced thromboembolic events, no consistent patterns were observed to suggest an association between carriage of specific *CYP2C9*, *CYP2C19*, *or CYP3A* variants and an increased frequency of complications. DVT/PE cases occurred both in normal metabolizers (NM) and in carriers of altered phenotypes (IM, PM), with no statistically significant differences between groups. Similarly, bleeding episodes were observed in patients with different *CYP2C9* and *CYP2C19* phenotypes, which also precluded the identification of gene-level associations (Table 11). These findings are most likely attributable to the limitations of our analysis, including the small sample size, the low frequency of clinical events, and the need for further studies in larger cohorts with extended and long-term follow-up.

On the other hand, similar analyses demonstrated that in the group of patients receiving 40 mg/day of DD217, carriage of the C–G–C–T haplotype (rs1045642–rs2032582–rs1128503–rs4148738) of the *ABCB1* gene was significantly more frequent among those who developed thromboembolic complications (*p* = 0.038; Table 12). This finding suggests that the haplotypic structure of *ABCB1*, rather than individual SNPs, may influence the risk of thrombotic outcomes.

However, at the level of individual markers, no significant associations with the incidence of adverse events were identified, as no differences were observed between groups (Table 13).

In the cohort of patients receiving 60 mg/day, no statistically significant differences were observed between those with and without adverse outcomes, either at the level of individual SNPs (Table 14) or *ABCB1* haplotypes (Table 15). The small number of bleeding and thrombotic events precluded more detailed analyses and limited the statistical power of the study.

## 3. Discussion

The inclusion of pharmacogenetic investigations in the clinical development of novel drugs at early stages of their life cycle represents an important direction in modern pharmaceutical research. Traditionally, the impact of genetic factors on pharmacokinetics and pharmacodynamics has been assessed only after market approval, once sufficient clinical experience has accumulated and interindividual differences in efficacy and safety have become evident. For DOACs such as rivaroxaban, apixaban, and dabigatran, meaningful insights into the contribution of cytochrome P450 and drug transporter gene polymorphisms to variability in pharmacological response were largely obtained retrospectively, under conditions of clinical use. Regulatory agencies, including the FDA and EMA, now strongly encourage the integration of pharmacogenetic approaches into the early stages of drug development and clinical trials, as this enables the identification of predictors of efficacy and risk of adverse events before a drug is widely released into the market [19,20,21]. Such an approach supports the development of a more comprehensive safety and efficacy profile at an early stage in the drug’s life cycle and can provide the foundation for algorithms guiding individualized pharmacotherapy. Experience with pharmacogenetic studies in phase I–II trials underscore their practical relevance: the identification of associations between ADME gene variants (*CYP2C9*, *CYP2C19*, *ABCB1*, among others) and pharmacokinetic parameters can inform the design of subsequent studies and reduce the likelihood of unfavorable outcomes [22,23].

The in silico screening confirmed the anticipated anticoagulant activity of DD217 and its inhibition of factor Xa (PASS: Pa = 0.894 and Pa = 0.609, respectively), providing internal consistency with preclinical and clinical evidence on the drug’s mechanism of action [24]. Beyond the pharmacodynamic profile, in silico tools suggested possible involvement of CYP2C family isoenzymes in DD217 metabolism: PASS predicted CYP2C8 inhibition, while results from multiple platforms (SwissADME, CYPstrate, P450-Analyzer, CYPlebrity) indicated both substrate potential and inhibitory activity for CYP2C9, CYP2C19, and CYP3A4. On the one hand, these dual signals supported the rationale for including CYP2C9, CYP2C19, CYP3A4, and CYP2C8 in the candidate genotyping panel, demonstrating how in silico results can serve as a prioritization tool for ADME targets. They also point to the theoretical possibility of clinically relevant drug–drug interactions (DDIs) when co-administered with substrates of these enzymes, warranting validation in in vitro systems and clinical settings [25,26]. Another interpretation is that potential inhibition could lead to phenoconversion—that is, attenuation of genetically determined differences in enzyme activity due to pharmacological blockade [27]. In other words, CYP inhibition may “mask” genetically driven variability in enzyme function, thereby reducing the observable differences between carriers of distinct allelic variants.

An additional in silico finding was the structural comparison: among molecules with similar metabolic profiles, betrixaban emerged as the closest analogue (Tanimoto = 0.587; Hausdorff = 0.358). This is consistent with the known pharmacokinetics of betrixaban, characterized by predominant circulation of the unchanged parent compound in plasma and the formation of two inactive hydrolytic metabolites with minimal CYP involvement (<1% of metabolism via CYP) (DrugBank. Betrixaban (DB12364). Available online: https://go.drugbank.com/drugs/DB12364 (accessed on 12 August 2025)). Betrixaban is also recognized as a P-glycoprotein (ABCB1) substrate, with corresponding interactions when co-administered with P-gp inhibitors [28]. These parallels allowed us to extrapolate structurally informed expectations regarding the contribution of P-gp transport to DD217 disposition and justified the inclusion of *ABCB1* variants among the candidate genes, a rationale partially supported by our results: *ABCB1* polymorphisms were indeed associated with altered drug exposure.

In our study, statistically significant associations were identified in the 60 mg/day cohort between specific *ABCB1* variants (rs1045642, rs2032582, rs1128503, and rs4148738) and AUC_last_ and C_max_ (*p* < 0.05), but not T_max_ (*p* > 0.05). In the 40 mg/day group, no significant associations with pharmacokinetic parameters were observed, either at the level of individual SNPs or haplotypes. Similarly, haplotype-based analyses in the 60 mg/day group did not reveal statistically significant associations. This distribution of effects appears biologically plausible and aligns with the classical pharmacokinetics of P-gp substrates in the gastrointestinal tract. P-gp plays a key role in limiting substrate absorption; variability in efflux activity primarily affects the extent of absorption (exposure and peak concentration), whereas the rate of peak appearance (T_max_) is more often determined by solubility constraints, gastric emptying, and baseline permeability, making it less sensitive to moderate alterations in efflux activity [29].

The dose-dependent nature of the associations (observed at 60 mg but not at 40 mg) may reflect saturation or threshold-level involvement of the efflux transporter P-gp at higher drug concentrations, where transporter-mediated efflux becomes rate-limiting and interindividual differences in its activity translate more strongly into variability in exposure. This interpretation is consistent with findings for other DOACs—particularly rivaroxaban and apixaban, both P-gp substrates. For rivaroxaban, *ABCB1* involvement has been demonstrated in cellular models, although the influence of specific *ABCB1* polymorphisms on transport has not always been observed in vitro [30], and clinical associations between *ABCB1* variants and DOAC exposure have not been consistently reported across studies, including population PK models and systematic reviews [8,9,31]. The heterogeneity of these findings may partly reflect the common practice in pharmacogenetic studies of “normalizing” concentrations relative to DOAC dose (C/Dose), which could interfere with or obscure potential marker effects. This issue warrants further investigation.

Three major *ABCB1* variants—3435T>C, 2677T>G/A, and 1236T>C—have been extensively studied; together they form a haplotype associated with altered P-gp function and, more broadly, with reduced activity and substrate specificity of this transporter [32]. However, evidence regarding their impact remains inconsistent. In our study, the absence of haplotype-level associations despite the presence of SNP-level effects is likely attributable to the low frequency of certain allele combinations and the limited sample size, which reduced statistical power, as well as to the possible functional nonequivalence of individual variants within haplotypes [11]. Moreover, only moderate linkage disequilibrium between these loci (r^2^ = 0.3–0.7 in European populations) suggests that their combined evaluation may have diluted the effects observed for individual SNPs.

For CYP2C9, an association between metabolic phenotype and T_max_ was observed at the 60 mg/day dose: carriers of reduced-function alleles (IM + PM) exhibited a significantly shorter time to reach C_max_ compared with normal metabolizers (NM), alongside a trend toward higher AUC_last_ and C_max_ values. At the 40 mg dose, no consistent pattern was found, which may reflect a combination of dose-dependent pharmacokinetic effects and limited subgroup size, whereas at 60 mg partial saturation of CYP2C9-mediated clearance could enhance differences in exposure.

The observed association between CYP2C9 phenotype and shorter T_max_ values in 60 mg group may be explained by reduced early-phase clearance and attenuated presystemic metabolism, resulting in more rapid and pronounced plasma concentration increases (C_max_) together with a tendency toward greater overall exposure (AUC). Although this shift could theoretically lead to an earlier onset of anticoagulant action, no consistent relationship with thrombotic or bleeding events was observed in our cohort. Further studies including time-matched anti-Xa pharmacokinetic/pharmacodynamic profiling and larger patient samples are warranted to clarify whether earlier peak exposure translates into clinically meaningful differences in DD217’s anticoagulant efficacy or safety.

Taken together, the results suggest that at the higher DD217 dose, intestinal efflux (ABCB1) and early elimination processes (CYP2C9) become rate-limiting steps, translating into differences in AUC and C_max_ for ABCB1 and in T_max_ for CYP2C9. At the lower dose (40 mg), the contribution of these pathways either does not limit systemic exposure or is masked by overall variability and the limited statistical power of the sample. This interpretation is consistent with fundamental pharmacokinetic and biopharmaceutical models describing the impact of P-gp and metabolic enzymes on absorption profiles and early-phase plasma concentrations [29]. Comparable findings have been reported for other DOACs: the evidence for CYP polymorphisms influencing exposure to factor Xa inhibitors is heterogeneous and generally weaker than for transporters. Systematic reviews and GWAS studies more consistently emphasize the role of ABCB1 and ABCG2 [8,33,34], whereas associations with CYP polymorphisms are less frequent, less consistent, and often dependent on study design and cohort size [9].

No significant associations were identified between carriage of *CYP2C9*, *CYP2C19*, *CYP3A4/5*, *CYP2C8*, or *ABCB1* variants and the incidence of thromboembolic complications or bleeding events, regardless of the daily DD217 dose. Isolated cases of thrombosis and bleeding were observed both in carriers of “normal” phenotypes and in patients with reduced enzymatic or transporter activity, precluding statistically significant differences between groups. First, the low frequency of clinical outcomes combined with the limited sample size substantially reduced the statistical power of the analysis. Second, the occurrence of thrombotic and bleeding complications is inherently multifactorial: in addition to pharmacokinetic determinants (including enzyme and transporter genetics), risk is influenced by clinical and demographic characteristics, comorbidities, concomitant pharmacotherapy, and other factors [35,36]. This complexity complicates efforts to disentangle the contribution of individual genetic markers in a small-scale analytical study. Even when such variants affect pharmacokinetic parameters (AUC_last_, C_max_, T_max_), translating these differences directly into clinical outcomes is challenging within a limited cohort. A clinically observable effect on complication risk typically requires either a more pronounced modification of drug exposure or the presence of additional risk factors. Similar observations have been reported in studies and meta-analyses of rivaroxaban and apixaban: while ABCB1 polymorphisms have been linked to altered drug exposure [33,34,37], consistent associations with bleeding or thrombotic risk have not been demonstrated [8,10,11], or risk was shaped by a combination of clinical and laboratory factors [13].

Overall, this study identified specific associations between *ABCB1* and *CYP2C9* genetic variants and the pharmacokinetic parameters of DD217, highlighting their potential role in shaping interindividual variability in drug exposure. Although consistent links between pharmacogenetic markers and clinical outcomes could not be demonstrated within the limited cohort, the findings, supported by in silico predictions, provide a biologically plausible basis for further exploration of these pathways in larger patient populations. The magnitude of the observed genotype effects appears modest and comparable to that reported for other DOACs, supporting the feasibility of early pharmacogenetic evaluation rather than indicating immediate clinical applicability.

### Limitations

This study has several limitations that must be considered when interpreting the findings. First, the limited sample size substantially reduced the statistical power of the analysis and the likelihood of detecting weaker associations. In addition, the low frequency of clinical outcomes (seven thrombotic events and two bleeding episodes) precluded a reliable assessment of the influence of genetic factors on the risk of complications, consistent with the limitations observed in other pharmacogenetic studies of DOACs. Another limitation is the relatively short observation period (14 days of therapy), which did not allow for the evaluation of long-term thrombotic and hemorrhagic events. Furthermore, the selection of genes was based on a candidate gene approach and in silico predictions, which does not permit the identification of unexpected associations that could emerge from genome-wide analyses. Finally, the results should be interpreted with caution due to the absence of in vitro verification of the predicted interactions (e.g., CYP2C9/2C19 inhibition, P-gp transport) and the limited generalizability of the findings to broader patient populations with diverse comorbidities. These limitations underscore the need for larger and longer-term studies employing multifactorial models that integrate both genetic and clinical determinants of risk—including demographic characteristics (age, sex), renal function, and concomitant medications—in line with current trends in anticoagulant pharmacogenetics [14,38].

## 4. Materials and Methods

### 4.1. Study Population

The design, eligibility criteria, and outcomes of the phase II clinical trial evaluating the safety and tolerability of DD217 (NCT05189002, ClinicalTrials.gov) have been described previously [39]. Briefly, this was a multicenter, double-blind, randomized, prospective phase II study designed to determine the optimal dosing regimen and to assess the safety and efficacy of DD217 for the prevention of venous thromboembolism (VTE) in patients undergoing knee replacement surgery. Patients were randomized into one of three groups: two groups received oral DD217 at doses of 40 mg or 60 mg once daily, starting on the morning of the first postoperative day, and one group received the active comparator, dalteparin sodium. The treatment duration was 14 days.

The study was conducted in compliance with Russian legislation and international regulatory standards, including the Declaration of Helsinki (World Medical Association, 2013) and the principles of Good Clinical Practice (National Standard of the Russian Federation, GOST R 52379-2005) [40]. All patients provided written informed consent prior to participation. The study protocol and related documents were approved by the Ethics Committee of the Ministry of Health of the Russian Federation (protocol excerpt No. 192, 21 May 2019). For the present analysis, blood samples and de-identified clinical and pharmacokinetic data were provided by the trial sponsor (Avexima Diol LLC, Vladivostok, Russia). All data were de-identified prior to analysis, and investigators had no access to personal identifiers. The authors of this manuscript were not involved in patient recruitment or in the conduct of the clinical trial.

In total, data from 52 patients who underwent knee replacement surgery were included in the analysis. Of these, 34 patients received DD217 at daily doses of 40 mg (*n* = 16) or 60 mg (*n* = 18), and 18 patients received dalteparin sodium. The clinical and demographic characteristics, as well as laboratory parameters of the study participants, are summarized in Table 16.

In total, seven cases of DVT/PE were documented during the trial: four in the DD217 40 mg/day group, one in the DD217 60 mg/day group, and two in the dalteparin sodium group. In addition, two bleeding events were recorded: one clinically relevant non-major bleed in the DD217 60 mg/day group and one clinically non-relevant minor bleed in the dalteparin sodium group. These data were subsequently used to evaluate the contribution of the studied genetic markers to the incidence of adverse events associated with DD217 therapy.

To characterize the pharmacokinetic profile of DD217 in our study, three parameters were assessed: C_max_, T_max_, and AUC_last_. The results are presented in Table 17.

### 4.2. In Silico Assessment of Pharmacological Potential of DD217

The structural formula of DD217 (*N*-(5-chloropyridin-2-yl)-5-methyl-2-[4-(*N*-methylacetimidamido)benzamido]benzamide hydrochloride) is shown in Figure 1.

The pharmacological potential of DD217 was assessed by predicting its effects, metabolism, and transporter interactions. For pharmacological effect prediction, we used the PASS program [41,42], a well-established program trained on large, curated structure–activity datasets. PASS was selected because of its broad coverage of pharmacological endpoints, repeated peer-reviewed validation, and suitability for exploring novel chemotypes. (see, e.g., Abdul-Hammed M. et al. (2022) [43]; Bocharova O.A. et al. (2023) [44]; Gangwal A. et al. (2025) [45]; Medvedeva S.M. et al. (2024) [46]; Muratov E.N. et al. (2020) [47]; Panina S.V. et al. (2022) [48]; Schimunek J. et al. (2024) [49]; Sukhachev V.S. et al. (2024) [50]). We used PASS 2022 which is based on the training set including over 1.6 million biologically active compounds. It predicts more than eight thousand biological activities with average accuracy of 0.936.

As a result of the prediction, the user obtains a list of probable types of activity with two values: Pa and Pi. Pa (Pi) represents the probability of the compound belonging to class of active (inactive) compounds, respectively.

Pharmacological effects and mechanisms of action predicted for DD217 with threshold at Pa > 0.5 are given in Table 18.

As one may see from Table 18, probable biological activities correspond to the pharmacological effects and mechanisms of action confirmed in the preclinical and clinical studies [23].

The result of the prediction of DD217’s interaction with metabolic enzymes, obtained using the PASS at the threshold Pa > Pi, is presented in Table 19.

It should be noted that CYP2C29 is a cytochrome P450 enzyme in mice, and CYP2C3 is in rabbits. CYP2C8 is a human enzyme.

For pharmacokinetic profiling of DD217 we used various widely issued publicly available online resources, including SwissADME [51], CYPstrate [52], P450 Analyzer [53], CYPlebrity [54], ADMETlab 2.0 [55] and MetaPASS [56]. These tools were chosen based on their predictive performance and their complementarity in terms of metabolic and transporter-related endpoints. The use of multiple prediction tools improves the reliability of the final assessment of the pharmacokinetic profile of DD217.

SwissADME and ADMETlab 2.0 provide a comprehensive analysis of ADME parameters and use multiple machine learning models to predict various parameters. They were used for assessing the belonging of DD217 to the inhibitors of cytochrome P450 enzymes and to inhibitors and substrates of P-gp. The CYPstrate resource is a tool focused on the prediction of substrates of major CYPs. It returns “Non-substrate” or “Substrate” values without specifying the probabilities of this value. The P450 Analyzer resource is focused on P450 enzymes interaction and provides both the estimation of pIC50 value (calculated using the GUSAR algorithm [57]) and the Pa and Pi values. CYPlebrity uses a high-quality ML algorithm to estimate the probability of inhibitory activity against P450.

The summary tables below (Table 20, Table 21 and Table 22) list the resources and prediction results for DD217 in relation to the substrates (Table 20 and Table 22) or inhibitors (Table 21 and Table 22) of the corresponding enzymes (CYP P450 or P-gp). For the P450 Analyzer resource, the Pa-Pi values are presented in the tables. The results of other resources are presented “as is”, including the phrase “No prediction” in CYPstrate web resource.

Based on the prediction results given in Table 20 and Table 21, it can be assumed that DD217 is the potential substrate and inhibitor of CYP2C.

Using the MetaPASS web resource [56], structures of the pharmaceutical substances similar to DD217 were found among the known metabolic schemes. The most similar was the direct oral anticoagulant Betrixaban, an inhibitor of factor X (DrugBank. Betrixaban (DB12364). Available online: https://go.drugbank.com/drugs/DB12364 (accessed on 12 August 2025)). The similarity between DD217 and Betrixaban is equal to 0.587 (Tanimoto coefficient, based on MNA descriptors) and 0.358 (Hausdorff distance, based on QNA descriptors). It is known [58] that unmetabolized Betrixaban is the predominant form in human plasma, followed by two hydrolytic CYP-independent inactive metabolites.

### 4.3. Candidate Gene Selection

The official drug label indicates that DD217 is metabolized to form a hydroxylated derivative. It also specifies a potential risk of drug-drug interactions at the level of cytochrome P450 enzymes, particularly CYP2C9 (e.g., losartan, diclofenac, ibuprofen, naproxen) and CYP3A4 (azole antifungals such as ketoconazole and fluconazole). Considering known metabolic pathways and the spectrum of predicted biological activities, DD217 most closely resembles the anticoagulant betrixaban, which is not approved in Russia.

Based on the predicted metabolic pathways from in silico modeling (derived from the structure of the active substance and its activity spectrum), available information on drug metabolism and potential drug–drug interactions, pharmacogenetic data from other DOACs, as well as a review of the scientific literature and the PharmGKB database (PharmGKB. Available online: https://www.pharmgkb.org/ (accessed on 15 April 2025)), we selected genetic markers that may potentially be associated with variability in the pharmacological response and pharmacokinetic parameters of DD217. A candidate gene panel was assembled, including the following single nucleotide polymorphisms (SNPs): *CYP2C9*2* (rs179985, 430C>T), *CYP2C9*3* (rs1057910, 1075A>C), *CYP2C19*2* (rs4244285, 681G>A), *CYP2C19*3* (rs4986893, 636G>A), *CYP2C19*17* (rs12248560, −806C>T), *CYP2C8*3* (rs10509681, T>C), *CYP2C8*3* (rs11572080, C>T), *CYP3A4*18B* (rs28371759, 878T>C), *CYP3A4*1B* (rs2740574, C>T), *CYP3A4*22* (rs35599367, C>T), *CYP3A5*3* (rs776746, A>G), and four variants of ABCB1 (rs4148738, C>T; rs1045642, 3435T>C; rs2032582, 2677G>T; rs1128503, 1236C>T). Selection of candidate genes for genotyping was guided by several criteria: the strength of available evidence, allele frequencies in the general population, and inclusion of the variants in international professional pharmacogenetic guidelines issued by the Clinical Pharmacogenetics Implementation Consortium, the Dutch Pharmacogenetics Working Group, and the Association for Molecular Pathology.

### 4.4. Genotyping

Whole blood was collected from the cubital vein into vacutainers containing 3.2% sodium citrate (Minimed LLC, Bryansk, Russia). After centrifugation, blood cells were isolated and used as the source of DNA for subsequent genotyping. Samples were stored at −80 °C until analysis.

Genomic DNA was extracted from blood cells using a sorbent-based DNA extraction kit (Syntol LLC, Moscow, Russia) according to the manufacturer’s instructions. DNA yield and purity were assessed with a NanoDrop 2000 spectrophotometer (Thermo Fisher Scientific, Waltham, MA, USA). Genotyping was performed using real-time PCR on a CFX96 Touch Real-Time System with CFX Manager software v3.0 (Bio-Rad, Hercules, CA, USA).

Genotyping was performed using commercial reagent kits with allele-specific probes (Syntol LLC, Moscow, Russia) used according to the manufacturer’s instructions. Each reaction mixture contained 10 μL PCR mix, 10 μL diluent, 0.5 μL Taq polymerase, and 5 μL genomic DNA from the tested sample. The amplification program for all polymorphisms, except CYP3A5 rs776746 and CYP2C19 rs12248560, consisted of an initial incubation at 95 °C for 3 min, followed by denaturation at 95 °C for 15 s and annealing at 63 °C for 40 s, repeated for 40 cycles, according to the manufacturer’s instructions. Fluorescence signals were detected in the corresponding channels: FAM and HEX. The amplification program included an initial incubation at 95 °C for 3 min, followed by denaturation at 95 °C for 15 s and annealing at 60 °C for 40 s, repeated for 40 cycles, in accordance with the manufacturer’s instructions. Fluorescence signals were detected in the following channels: HEX and ROX for CYP3A5 rs776746, and FAM and HEX for CYP2C19 rs12248560.

Genotyping of CYP2C8 (rs10509681, rs11572080) and CYP3A4 (rs28371759) was performed using commercial kits “TaqMan^®^ SNP Genotyping Assays” and TaqMan Universal Master Mix II, no UNG (Thermo Fisher Scientific, Waltham, MA, USA) according to the manufacturer’s instructions. Each reaction contained 0.5 μL of “TaqMan^®^ SNP Genotyping Assay” (diluted 1:40), 10 μL of TaqMan Universal Master Mix II, 9.5 μL of RNase-free water, and 5 μL of genomic DNA from the tested sample. The amplification program consisted of an initial incubation at 95 °C for 10 min, followed by denaturation at 95 °C for 15 s and annealing at 60 °C for 1 min, repeated for 50 cycles. Fluorescence signals were detected in the FAM and VIC channels.

Genotyping of CYP3A4 rs35599367 and ABCB1 rs4148738 was performed using commercial reagent kits with allele-specific probes (TestGen LLC, Ulyanovsk, Russia), according to the manufacturer’s instructions. Each reaction mixture contained 4 μL PCR mix, 2 μL Taq polymerase, 3 μL water, and 1 μL genomic DNA from the tested sample. The amplification program consisted of an initial incubation at 95 °C for 2 min, followed by denaturation at 94 °C for 10 s and annealing at 60 °C for 30 s, repeated for 40–50 cycles. Fluorescence signals were detected in the FAM and HEX channels.

For each of the three genotyping platforms, every analytical run included positive controls (reference samples with known genotypes provided by the manufacturer) and negative controls (no-template controls) to exclude contamination. Approximately 10% of study samples were genotyped in duplicate, yielding a 100% concordance rate. Samples with low fluorescence signal intensity or ambiguous clustering were reanalyzed or excluded from the final dataset. DNA yield and purity were confirmed before analysis (OD260/280 ratio 1.8–2.0).

### 4.5. Statistical Analysis

Study data were analyzed using both parametric and non-parametric statistical methods. All analyses were performed with Statsoft Statistica 12.0 (Dell Statistica, Tulsa, OK, USA).

Quantitative variables were assessed for normality of distribution using the Kolmogorov–Smirnov test as well as measures of skewness and kurtosis. For variables demonstrating a normal distribution, data were summarized as arithmetic means (M) and standard deviations (SD).

For comparisons of mean values in normally distributed datasets, Student’s *t*-test was applied. The obtained t-values were evaluated against critical values, with differences considered statistically significant at *p* < 0.05.

For comparisons involving multiple groups of quantitative data that did not follow a normal distribution, the Kruskal–Wallis test was used as a non-parametric alternative to one-way ANOVA. The Kruskal–Wallis statistic was calculated after ranking all elements of the analyzed datasets. Bonferroni correction was applied to account for multiple testing. If the calculated value exceeded the critical threshold, differences were deemed statistically significant; otherwise, the null hypothesis was accepted.

When statistically significant differences between groups were identified, pairwise comparisons were additionally performed using Dunn’s post hoc test.

## 5. Conclusions

This study provides the first insights into the influence of genetic polymorphisms on the pharmacokinetics and clinical outcomes of DD217 therapy. Associations were identified between *CYP2C9* variants and changes in T_max_, as well as between *ABCB1* variants and drug exposure (AUC_last_, C_max_), findings that are consistent with in silico predictions and biologically plausible given the role of CYP2C enzymes and the P-gp transporter in drug metabolism and absorption. At the same time, no consistent associations with the incidence of thrombotic events or bleeding were observed, warranting cautious interpretation of these results. The limited sample size, low frequency of clinical events, and short observation period preclude definitive conclusions regarding the clinical relevance of the detected associations. Nevertheless, this work illustrates the potential value of incorporating pharmacogenetic studies early in the drug life cycle. Such investigations are rarely conducted at this stage of development, yet they can provide the foundation for personalized medicine and improved treatment safety. Future research should focus on larger patient cohorts, longer follow-up periods, and the use of genome-wide analyses to explore pharmacokinetic and pharmacodynamic correlations, as well as the construction of multifactorial models that integrate clinical and demographic risk factors. We believe that such efforts will contribute to building the evidence base for the role of pharmacogenetics in personalizing DOAC therapy and may ultimately lead to the development of practical dosing algorithms based on a patient’s genetic profile.

## Figures and Tables

**Figure 1 pharmaceuticals-18-01617-f001:**
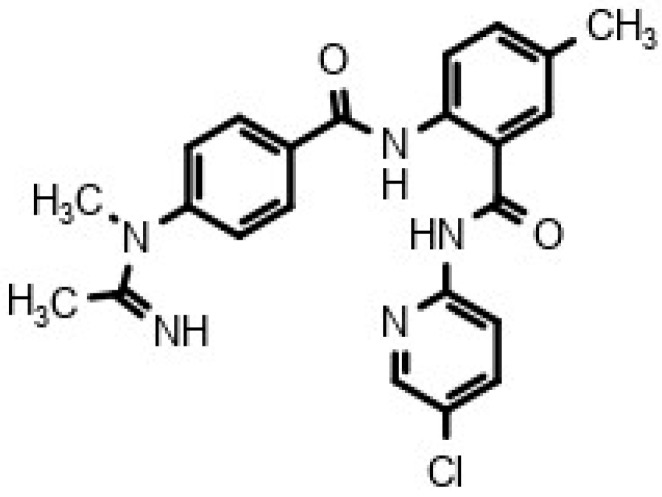
Chemical structure of DD217.

**Table 1 pharmaceuticals-18-01617-t001:** Distribution of genetic variants in the study cohort.

Gene	SNP	Frequency			Minor Allele Frequency (%)	HWE Conformity	
		Genotype	Observed (*n*)	Expected (*n*)		χ^2^	*p*-Value
*CYP2C9*	rs1799853	CC	41	41.58	10.6	0.727	0.695
		CT	11	9.84			
		TT	0	0.58			
	rs1057910	AA	42	41.58	10.6	0.376	0.829
		AC	9	9.84			
		CC	1	0.58			
*CYP2C19*	rs4244285	GG	44	43.39	8.7	1.147	0.563
		GA	7	8.22			
		AA	1	0.39			
	rs4986893	GG	51	51.00	1.0	0.005	0.998
		GA	1	0.99			
		AA	0	0.00			
	12248560	CC	29	29.25	25.0	0.034	0.983
		CT	20	19.50			
		TT	3	3.25			
*CYP2C8*	rs10509681	TT	45	45.24	6.7	0.271	0.873
		TC	7	6.53			
		CC	0	0.24			
	rs11572080	CC	45	45.24	6.7	0.271	0.873
		CT	7	6.53			
		TT	0	0.24			
*CYP3A4*	rs35599367	CC	50	50.02	1.9	0.020	0.990
		CT	2	1.96			
		TT	0	0.02			
	rs28371759	AA	52	NA	0	-	-
		AG	0	NA			
		GG	0	NA			
	rs2740574	AA	48	48.08	3.8	0.083	0.959
		AG	4	3.85			
		GG	0	0.08			
*CYP3A5*	rs776746	GG	45	45.24	6.7	0.271	0.873
		GA	7	6.53			
		AA	0	0.24			
*ABCB1*	rs1045642	TT	12	12.50	51.0	0.078	0.962
		TC	27	25.99			
		CC	13	13.50			
	rs2032582	GG	19	17.89	41.3	0.403	0.817
		GT	23	25.22			
		TT	10	8.89			
	rs1128503	TT	19	19.08	39.4	0.002	0.999
		TC	25	24.84			
		CC	8	8.08			
	rs4148738	CC	10	8.89	58.7	0.403	0.817
		CT	23	25.22			
		TT	19	17.89			

**Table 2 pharmaceuticals-18-01617-t002:** Distribution of CYP phenotypic categories among study participants receiving DD217.

Cytochrome	Phenotype	40 mg (*n*, %)	60 mg (*n*, %)	Genotypes	*p*-Value
CYP2C9	PM	2 (12.5)	2 (11.1)	*2/*3, *3/*3	0.884
	IM	6 (37.5)	5 (27.8)	*1/*3, *1/*2	0.765
	NM	8 (50)	11 (61.1)	*1/*1	0.491
CYP2C19	PM	-	1 (5.6)	*2/*2	NA
	IM	1 (6.25)	3 (16.7)	*1/*2, *2/*17	0.722
	NM	6 (37.5)	8 (44.4)	*1/*1	0.287
	RM	7 (43.75)	6 (33.3)	*1/*17	1.000
	UM	2 (12.5)	-	*17/*17	NA
CYP3A	IM	12 (75)	16 (88.9)	*1/*1 + *3/*3 *1/*22 + *1/*3	0.132
	EM	4 (25)	2 (11.1)	*1/*1 + *1/*3 *1/*1 + *1/*1	0.378

Note: PM—poor metabolizers; IM—intermediate metabolizers; NM—normal metabolizers; RM—rapid metabolizers; UM—ultrarapid metabolizers; EM—extensive metabolizers.

**Table 3 pharmaceuticals-18-01617-t003:** Association of *CYP2C9*, *CYP2C19*, and *CYP3A* variants with T_max_, AUC_last_, and C_max_ of DD217.

Cytochrome	DD217 Dose	Genotype	Phenotype	T_max_	AUC_last_	C_max_	*p*-Value
CYP2C9	40 mg	*1/*1	NM (*n* = 8)	8.00 ± 7.15	54.50 ± 49.37	7.28 ± 8.71	>0.05 T_max_: *p* _NM vs. IM+PM_ = 0.3894 AUC: *p* _NM vs. IM+PM_ = 0.2786 C_max_: *p* _NM vs. IM+PM_ = 0.2345
		*1/*2, *1/*3	IM (*n* = 6)	9.00 ± 8.17	33.86 ± 36.25	3.72 ± 5.61	
		*2/*3, *3/*3	PM (*n* = 2)	24.0	18.65 ± 22.50	3.11 ± 3.75	
	60 mg	*1/*1	NM (*n* = 11)	12.91 ± 9.14	37.55 ± 42.41	3.66 ± 4.07	0.002 * T_max_: *p* = 0.005227 * AUC: *p* _NM vs. IM+PM_ = 0.06926 C_max_: *p* _NM vs. IM+PM_ = 0.1259
		*1/*2, *1/*3	IM (*n* = 5)	3.20 ± 2.17	89.75 ± 89.27	9.24 ± 9.59	
		*2/*3	PM (*n* = 2)	2.50 ± 2.12	177.19 ± 223.61	11.28 ± 13.60	
CYP2C19	40 mg	*1/*17, *17/*17	RM + UM (*n* = 9)	7.67 ± 6.71	47.70 ± 42.56	5.80 ± 6.73	>0.05 T_max_: *p* _RM+UM vs. NM vs. IM+PM_ = 0.2123 AUC: *p* _RM+UM vs. NM vs. IM+PM_ = 0.3765 C_max_: *p* _RM+UM vs. NM vs. IM+PM_ = 0.6185
		*1/*1	NM (*n* = 6)	15.50 ± 9.95	33.13 ± 48.12	5.25 ± 8.82	
		*1/*2	IM (*n* = 1)	4.00	48.45	3.12	
	60 mg	*1/*17	RM (*n* = 6)	9.17 ± 7.33	50.5 ± 51.61	4.50 ± 4.61	>0.05 T_max_: *p* _RM vs. NM vs. IM+PM_ = 0.5459 AUC: *p* _RM vs. NM vs. IM+PM_ = 0.8896 C_max_: *p* _RM vs. NM vs. IM+PM_ = 0.806
		*1/*1	NM (*n* = 8)	8.75 ± 10.07	98.0 ± 123.81	8.44 ± 9.71	
		*1/*2, *2/*2, *2/*17	IM + PM (*n* = 4)	9.50 ± 9.98	32.28 ± 22.5	3.64 ± 3.51	
CYP3A	40 mg	*1/*1	EM (*n* = 12)	11.75 ± 9.50	40.04 ± 42.11	5.02 ± 6.92	>0.05 T_max_: *p* _EM vs. IM_ = 0.5345 AUC: *p* _EM vs. IM_ = 0.5989 C_max_: *p* _EM vs. IM_ = 0.5209
		*1/*22, *1/*3	IM (*n* = 4)	6.25 ± 3.50	48.99 ± 48.74	6.63 ± 8.63	
	60 mg	*1/*1	EM (*n* = 16)	9.75 ± 8.93	52.71 ± 63.32	5.32 ± 6.58	>0.05 T_max_: *p* _EM vs. IM_ = 0.3201 AUC: *p* _EM vs. IM_ = 0.2092 C_max_: *p* _EM vs. IM_ = 0.3268
		*1/*3	IM (*n* = 2)	3.50 ± 3.54	186.36 ± 210.65	11.94 ± 12.67	

Note: *—Statistically significant *p*-values were observed; RM—rapid metabolizers; EM—extensive metabolizers; NM—normal metabolizers; IM—intermediate metabolizers; PM—poor metabolizers.

**Table 4 pharmaceuticals-18-01617-t004:** Association of *CYP2C8* variants with T_max_, AUC_last_, and C_max_ of DD217.

Group	SNP	Genotype	*n*	T_max_ (h, SD)	*p*-Value (T_max_)	AUC_last_ (SD)	*p*-Value (AUC_last_)	C_max_ (SD)	*p*-Value (C_max_)
40 mg	rs10509681	T/C	3	16.67 (7.33)	0.17	46.88 (28.46)	0.84	7.06 (4.27)	0.67
		T/T	13	8.92 (2.04)		41.22 (11.86)		5.05 (2.02)	
	rs11572080	C/C	13	8.92 (2.04)	0.17	41.22 (11.86)	0.84	5.05 (2.02)	0.67
		C/T	3	16.67 (7.33)		46.88 (28.46)		7.06 (4.27)	
60 mg	rs10509681	T/C	2	3 (1)	0.31	62.47 (43.39)	0.94	5.69 (4.02)	0.94
		T/T	16	9.81 (2.23)		68.2 (23.49)		6.11 (1.89)	
	rs11572080	C/C	16	9.81 (2.23)	0.31	68.2 (23.49)	0.94	6.11 (1.89)	0.94
		C/T	2	3 (1)		62.47 (43.39)		5.69 (4.02)	

Note: T_max_—time to maximum plasma concentration; AUC_last_—area under the concentration-time curve up to the last measurable point; C_max_—maximum plasma concentration.

**Table 5 pharmaceuticals-18-01617-t005:** Distribution of *CYP2C8* haplotypes and their association with pharmacokinetic parameters of DD217.

Group	rs10509681	rs11572080	Haplotype Frequency	*p*-Value		
				T_max_	AUC_last_	C_max_
40 mg/day	T	C	0.9062	0.17	0.84	0.67
	C	T	0.0938			
60 mg/day	T	C	0.9444	0.31	0.94	0.94
	C	T	0.0556			

**Table 6 pharmaceuticals-18-01617-t006:** Association of *ABCB1* variants with T_max_, AUC_last_, and C_max_ of DD217 in patients receiving 40 mg/day.

SNP	Genotype	*n*	T_max_ (SD)	*p*-Value (T_max_)	AUC_last_ (SD)	*p*-Value (AUC_last_)	C_max_ (SD)	*p*-Value (C_max_)
rs1045642 T>C	T/T	4	9.5 (4.99)	0.074	35.53 (22.23)	0.17	4.63 (3.47)	0.42
	T/C	7	6 (1.56)		63.6 (18.31)		8.03 (3.45)	
	C/C	5	17.2 (4.18		17.83 (5.08)		2.42 (1)	
rs2032582 G>T	G/G	8	12.88 (3.44)	0.36	34.57 (14.34)	0.73	4.84 (2.64)	0.84
	G/T	6	9.5 (3.12)		46.36 (19.01)		5.24 (2.94)	
	T/T	2	3 (1)		60.88 (40.28)		8.35 (6.66)	
rs1128503 T>C	C/C	8	12.88 (3.44)	0.36	34.57 (14.34)	0.73	4.84 (2.64)	0.84
	C/T	6	9.5 (3.12)		46.36 (19.01)		5.24 (2.94)	
	T/T	2	3 (1)		60.88 (40.28)		8.35 (6.66)	
rs4148738 C>T	C/C	2	3 (1)	0.36	60.88 (40.28)	0.73	8.35 (6.66)	0.84
	T/C	6	9.5 (3.12)		46.36 (19.01)		5.24 (2.94)	
	T/T	8	12.88 (3.44)		34.57 (14.34)		4.84 (2.64)	

Note: T_max_—time to maximum plasma concentration; AUC_last_—area under the concentration-time curve up to the last measurable point; C_max_—maximum plasma concentration.

**Table 7 pharmaceuticals-18-01617-t007:** Distribution of *ABCB1* haplotypes in patients receiving DD217 40 mg/day and their association with pharmacokinetic parameters.

#	rs1045642	rs2032582	rs1128503	rs4148738	Frequency	*p*-Value		
						T_max_	AUC_last_	C_max_
1	C	G	C	T	0.5312	0.61	0.9	0.84
2	T	T	T	C	0.3125			
3	T	G	C	T	0.1563			
4	C	T	T	C	0			

**Table 8 pharmaceuticals-18-01617-t008:** Association of *ABCB1* variants with T_max_, AUC_last_, and C_max_ of DD217 in patients receiving 60 mg/day.

SNP	Genotype	*n*	T_max_ (SD)	*p*-Value (T_max_)	AUC_last_ (SD)	*p*-Value (AUC_last_)	C_max_ (SD)	*p*-Value (C_max_)
rs1045642	T/T	7	13.57 (3.72)	0.13	27.6 (5.59)	0.0094 *	2.42 (0.56)	0.013 *
	T/C	8	7.75 (2.63)		53.88 (18.59)		5.54 (1.74)	
	C/C	3	2 (1)		197.3 (93.48)		15.95 (7.25)	
rs2032582	G/G	4	2 (0.71)	0.12	163.65 (74.17)	0.03 *	14.13 (5.44)	0.018 *
	T/G	9	9.56 (2.82)		51.62 (16.19)		4.87 (1.49)	
	T/T	5	13.8 (4.39)		19.38 (5.28)		1.74 (0.51)	
rs1128503	T/T	4	16.25 (4.7)	0.057	14 (4.32)	0.029 *	1.49 (0.41)	0.019 *
	T/C	10	9 (2.59)		50.55 (14.43)		4.66 (1.36)	
	C/C	4	2 (0.71)		163.65 (74.17)		14.13 (5.44)	
rs4148738	C/C	5	10.6 (3.63)	0.18	18.86 (5.38)	0.03 *	1.78 (0.5)	0.019 *
	C/T	9	11.33 (3.23)		51.92 (16.11)		4.85 (1.5)	
	T/T	4	2 (0.71)		163.65 (74.17)		14.13 (5.44)	

Note: *—Statistically significant *p*-values were observed; T_max_—time to maximum plasma concentration; AUC_last_—area under the concentration-time curve up to the last measurable point; C_max_—maximum plasma concentration.

**Table 9 pharmaceuticals-18-01617-t009:** Distribution of *ABCB1* haplotypes in patients receiving DD217 60 mg/day and their association with pharmacokinetic parameters.

#	rs1045642	rs2032582	rs1128503	rs4148738	Frequency	*p*-Value		
						T_max_	AUC_last_	C_max_
1	T	T	T	C	0.4416	0.17	0.99	0.94
2	C	G	C	T	0.3021			
3	T	G	C	T	0.1139			
4	C	T	T	C	0.0306			
5	C	G	T	T	0.0284			
6	T	T	C	C	0.0284			
7	C	G	C	C	0.0278			
8	T	T	C	T	0.0278			

**Table 10 pharmaceuticals-18-01617-t010:** Genotypic and phenotypic profiles of patients with adverse events (DVT/PE or bleeding).

Adverse Event	Patient ID	DD217 Dose	Phenotype			CYP2C8*3		ABCB1			
			CYP2C9	CYP2C19	CYP3A	rs10509681	rs11572080	rs1045642	rs2032582	rs1128503	rs4148738
DVT/PE	112001	40 mg	NM	RM	EM	T	C	C	G	C	T
	113030	40 mg	NM	RM	EM	T	C	C	G	C	T
	113032	40 mg	PM	NM	EM	C	T	C	G	C	T
	113035	40 mg	NM	RM	IM	T	C	T	T	T	C
	113036	60 mg	NM	RM	EM	T	C	T	T	T	C
Bleeding	112008	60 mg	IM	NM	EM	T	C	C	G	C	T

Note: RM—rapid metabolizers; EM—extensive metabolizers; NM—normal metabolizers; IM—intermediate metabolizers; PM—poor metabolizers.

**Table 11 pharmaceuticals-18-01617-t011:** Association of *CYP2C9*, *CYP2C19*, and *CYP3A* variants with adverse events (DVT/PE or bleeding).

Cytochrome	DD217 Dose	Genotype	Phenotype	DVT/PE (*n*)	*p*-Value	Bleeding (*n*)	*p*-Value
CYP2C9	40 mg	*1/*1	NM (*n* = 8)	3	*p* _NM vs. IM+PM_ = 0.5692	0	NA
		*1/*2, *1/*3	IM (*n* = 6)	0		0	
		*2/*3, *3/*3	PM (*n* = 2)	1		0	
	60 mg	*1/*1	NM (*n* = 11)	1	*p* _NM vs. IM+PM_ = 1.0000	0	*p* _NM vs. IM+PM_ = 0.3889
		*1/*2, *1/*3	IM (*n* = 5)	0		1	
		*2/*3	PM (*n* = 2)	0		0	
CYP2C19	40 mg	*1/*17, *17/*17	RM + UM (*n* = 9)	3	*p* _RM+UM vs. NM vs. IM+PM_ = 0.5846	0	NA
		*1/*1	NM (*n* = 6)	1		0	
		*1/*2	IM (*n* = 1)	0		0	
	60 mg	*1/*17	RM (*n* = 6)	1	*p* _RM vs. NM vs. IM+PM_ = 0.3333	0	*p* _RM vs. NM+IM+PM_ = 0.3333
		*1/*1	NM (*n* = 8)	0		1	
		*1/*2, *2/*2, *2/*17	IM + PM (*n* = 4)	0		0	
CYP3A	40 mg	*1/*1	EM (*n* = 12)	3	*p* _EM vs. IM_ = 1.0000	0	NA
		*1/*22, *1/*3	IM (*n* = 4)	1		0	
	60 mg	*1/*1	EM (*n* = 16)	1	*p* _EM vs. IM_ = 1.0000	1	*p* _EM vs. IM_ = 1.0000
		*1/*3	IM (*n* = 2)	0		0	

Note: RM—rapid metabolizers; UM—ultrarapid metabolizers; EM—extensive metabolizers; NM—normal metabolizers; IM—intermediate metabolizers; PM—poor metabolizers.

**Table 12 pharmaceuticals-18-01617-t012:** Distribution of *ABCB1* haplotypes in patients receiving DD217 40 mg/day and their association with thromboembolic outcomes.

#	rs1045642	rs2032582	rs1128503	rs4148738	Frequency	*p*-Value
1	C	G	C	T	0.5312	0.038 *
2	T	T	T	C	0.3125	
3	T	G	C	T	0.1563	
4	C	T	T	C	0	

Note: *—Statistically significant *p*-values were observed.

**Table 13 pharmaceuticals-18-01617-t013:** Association of *ABCB1* variants with thromboembolic outcomes in patients receiving DD217 40 mg/day.

SNP	Genotype	*n*	Without DVT/PE	With DVT/PE	*p*-Value
rs1045642	T/T	4	4	0	0.063
	T/C	7	6	1	
	C/C	5	2	3	
rs2032582	G/G	8	5	3	0.37
	G/T	6	5	1	
	T/T	2	2	0	
rs1128503	C/C	8	5	3	0.37
	C/T	6	5	1	
	T/T	2	2	0	
rs4148738	C/C	2	2	0	0.37
	T/C	6	5	1	
	T/T	8	5	3	

Note: DVT/PE—deep vein thrombosis/pulmonary embolism.

**Table 14 pharmaceuticals-18-01617-t014:** Association of *ABCB1* variants with adverse outcomes in patients receiving DD217 60 mg/day.

SNP	Genotype	*n*	DVT/PE (*n*)	*p*-Value (DVT/PE)	Bleeding (*n*)	*p*-Value (Bleeding)
rs1045642	T/T	7	0	0.43	0	0.14
	T/C	8	1		0	
	C/C	3	0		1	
rs2032582	G/G	4	0	0.49	1	0.2
	G/T	9	1		0	
	T/T	5	0		0	
rs1128503	C/C	4	0	0.54	1	0.2
	C/T	10	1		0	
	T/T	4	0		0	
rs4148738	C/C	5	0	0.49	0	0.2
	T/C	9	1		0	
	T/T	4	0		1	

Note: DVT/PE—deep vein thrombosis/pulmonary embolism.

**Table 15 pharmaceuticals-18-01617-t015:** Distribution of *ABCB1* haplotypes in patients receiving DD217 60 mg/day and association with adverse outcomes.

#	rs1045642	rs2032582	rs1128503	rs4148738	Frequency	*p*-Value (DVT/PE)	*p*-Value (Bleeding)
1	T	T	T	C	0.441	0.98	0.79
2	C	G	C	T	0.3021		
3	T	G	C	T	0.1139		
4	C	T	T	C	0.0306		
5	C	G	T	T	0.0284		
6	T	T	C	C	0.0284		
7	C	G	C	C	0.0278		
8	T	T	C	T	0.0278		

Note: DVT/PE—deep vein thrombosis/pulmonary embolism.

**Table 16 pharmaceuticals-18-01617-t016:** General clinical and laboratory characteristics of study participants.

Parameter (Me ± SD/M/*n*)		In Total Cohort	DD217		Dalteparin Sodium
			40 mg/day	60 mg/day	
Sample size		52	16	18	18
Sex	Male	7	3	2	2
	Female	45	13	16	16
Age, years		63 ± 6.11	62.5 ± 6.02	62 ± 4.9	63 ± 7.48
BMI, kg/m^2^		34 ± 4.69	34 ± 3.41	34.3 ± 4.7	34.2 ± 5.78
Complete blood count and biochemical blood test					
Sodium, mmol/L		141	141	141.5	142
Potassium, mmol/L		4.21 ± 0.47	4.28 ± 0.33	4.32 ± 0.56	4.03 ± 0.45
Glucose, mmol/L		6.1	6.11	6.1	6.14
Total protein, g/L		72.44 ± 3.68	72.44 ± 3.29	73.06 ± 3.69	71.83 ± 4.09
Albumin, g/L		41.69 ± 2.37	41.68 ± 2.88	41.81 ± 2.33	41.59 ± 2.02
C-reactive protein, mg/L		2.65	2.05	3.1	3.25
Total cholesterol, mmol/L		6.05 ± 1.21	6.24 ± 1.46	5.82 ± 1.04	6.1 ± 1.16
Total bilirubin, µmol/L		11.1	11.25	11.5	10.8
Direct bilirubin, µmol/L		2.13 ± 0.68	2.08 ± 0.57	2.23 ± 0.69	2.08 ± 0.78
Indirect bilirubin, µmol/L		9.15	9	9.35	8.9
ALT, U/L		17	18	17.5	16.5
AST, U/L		20.5	20.5	20.5	20
GGT, U/L		23	21.5	23	20
Alkaline phosphatase, U/L		78.65 ± 22.57	72.81 ± 19.23	79.33 ± 24.91	83.17 ± 22.95
Lipase, U/L		24 ± 15.03	28 ± 17.81	24 ± 13.2	23 ± 14.35
Amylase, U/L		20.35 ± 8.94	20.9 ± 11.18	19.75 ± 8.78	21.1 ± 6.69
Creatinine, µmol/L		76.3 ± 12.05	79.85 ± 8.85	77.15 ± 16.45	72.35 ± 8.94
eGFR, mL/min/1.73 m^2^		74.47 ± 10.86	73.21 ± 10.92	73.42 ± 10.69	76.64 ± 11.26
Hematology					
RBC, ×10^12^/L		4.67 ± 0.41	4.6 ± 0.49	4.74 ± 0.43	4.65 ± 0.29
MCV, fL		90.35	90.85	90.8	89.95
MCH, pg		30.05	30.05	30	30.4
MCHC, g/dL		33.28 ± 0.67	33.35 ± 0.8	33.13 ± 0.58	33.37 ± 0.65
ESR, mm/h		16.71 ± 8.24	15.81 ± 6.73	17.22 ± 7.02	17 ± 10.66
Hematocrit, %		41.79 ± 3.31	41.73 ± 4.32	42.23 ± 2.98	41.42 ± 2.68
Hemoglobin, g/dL		13.91 ± 1.13	13.91 ± 1.52	13.97 ± 0.85	13.84 ± 1.05
Platelets, ×10^9^/L		248	240	259	250.5
WBC, ×10^9^/L		6.05	5.6	5.9	6.15
Neutrophils, %		57.75 ± 8.22	55.66 ± 8.99	60.23 ± 8.82	57.13 ± 6.53
Eosinophils, %		1.75	2.05	1.5	1.85
Basophils, %		1	1.2	1	0.9
Monocytes, %		5.95	5.6	5.65	6.2
Lymphocytes, %		29 ± 6.93	30.61 ± 7.47	26.36 ± 7.17	30.22 ± 5.66
Coagulation tests					
aPTT, sec		34.48 ± 3.48	35.29 ± 3.68	34.45 ± 3.83	33.8 ± 2.91
INR		0.97 ± 0.05	0.98 ± 0.06	0.96 ± 0.05	0.98 ± 0.05
D-dimer, ng/mL		455	462.5	359	508.5
Safety outcomes					
PE/DVT, *n*		7	4	1	2
Bleeding, *n*		2	0	1	1

**Table 17 pharmaceuticals-18-01617-t017:** Pharmacokinetic parameters (T_max_, AUC_last_, C_max_) of DD217 in the study population.

Subject ID	Dose, mg/day	Age	Sex	BMI	T_max_, h	AUC_last_	C_max_
112001	40	57	Woman	36.30	24	21.211	3.535
112013	40	54	Woman	34.00	24	2.735	0.456
112018	40	63	Woman	35.30	8	84.631	5.034
112020	40	65	Woman	30.50	12	10.412	0.72
112027	40	70	Man	28.70	24	4.928	0.405
112031	40	72	Woman	29.70	4	48.451	3.119
112034	40	63	Woman	38.90	8	17.602	1.379
112040	40	56	Man	37.90	8	39.7	3.95
113004	40	62	Woman	39.10	1	128.526	22.764
113014	40	58	Man	34.00	4	20.601	1.686
113016	40	61	Woman	37.20	8	13.049	1.136
113021	40	64	Woman	30.40	8	18.213	1.739
113030	40	71	Woman	31.50	6	10.234	0.656
113032	40	72	Woman	36.70	24	34.56	5.76
113033	40	55	Woman	31.60	2	101.166	15.011
113035	40	61	Woman	32.90	1	120.462	19.462
112003	60	64	Woman	35.90	4	19.08	1.663
112008	60	60	Woman	33.80	1	237.522	25.28
112010	60	56	Woman	34.70	4	35.852	3.324
112012	60	58	Man	34.70	24	16.597	1.155
112021	60	60	Woman	34.90	6	21.462	1.59
112022	60	71	Woman	25.70	2	105.854	9.707
112028	60	60	Woman	36.40	2	58.513	7.329
112037	60	57	Woman	39.00	12	4.589	0.325
112039	60	65	Woman	42.10	5	25.375	2.291
113006	60	51	Woman	30.10	8	51.929	4.963
113015	60	61	Woman	37.90	6	37.407	2.974
113020	60	60	Woman	34.60	24	14.483	1.614
113025	60	66	Woman	30.00	8	13.976	1.37
113029	60	68	Woman	34.00	2	62.698	8.694
113031	60	63	Woman	30.20	24	11.57	1.712
113036	60	63	Woman	33.10	6	152.359	13.569
113038	60	65	Man	22.00	24	11.552	0.613
113039	60	68	Woman	31.60	1	335.306	20.897

Note: T_max_—time to maximum plasma concentration; AUC_last_—area under the concentration-time curve up to the last measurable point; C_max_—maximum plasma concentration.

**Table 18 pharmaceuticals-18-01617-t018:** Prediction of potential pharmacological effects and mechanisms of action for DD217.

Pa	Pi	Activity
0.894	0.004	Anticoagulant
0.609	0.002	Factor Xa inhibitor

**Table 19 pharmaceuticals-18-01617-t019:** Prediction of the interaction of DD217 with metabolic enzymes.

Pa	Pi	Activity
0.229	0.109	CYP2C29 substrate
0.267	0.149	CYP2C8 inhibitor
0.185	0.144	CYP2C3 substrate

**Table 20 pharmaceuticals-18-01617-t020:** Prediction of DD217 belonging to the substrates of cytochrome P450 enzymes.

Web-Resource	1A2	2A6	2D6	2C8	2C19	2C9	3A4
CYPstrate (model “best performance”)	No prediction	Non-substrate	No prediction	Substrate	No prediction	Substrate	Substrate
CYPstrate (model “full coverage”)	Non-substrate	Non-substrate	Substrate	Substrate	Substrate	Substrate	Substrate
ADMETlab 2.0	0.917	No prediction	0.762	No prediction	0.079	0.213	0.600

**Table 21 pharmaceuticals-18-01617-t021:** Prediction of DD217 belonging to the inhibitors of cytochrome P450 enzymes.

Web-Resource	1A2	2D6	2C19	2C9	3A4
SwissADME	-	-	+	+	+
P450 Analyzer	0.115	−0.525	0.287	0.242	−0.628
CYPlebrity	0.58	0.28	0.56	0.45	0.42
ADMETlab 2.0	0.256	0.692	0.453	0.78	0.453

**Table 22 pharmaceuticals-18-01617-t022:** Prediction of DD217 belonging to the inhibitors/substrate of P-gp.

Web-Resource	Substrate	Inhibitor
SwissADME	-	
ADMETlab 2.0	0.996	0.851

## Data Availability

The datasets generated and analyzed during the current study are not publicly available due to privacy and ethical restrictions, as they contain clinical information obtained from participants of a sponsor-initiated clinical trial. De-identified data supporting the findings of this study are available from the corresponding author upon reasonable request and with permission of the trial sponsor (Avexima Diol LLC, Vladivistok, Russia). Requests should include a detailed research proposal and data protection plan.

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
