# Peer review of "An Investigational Study on the Role of ADME Agents’ Genetic Variation on DD217 Pharmacokinetics and Safety Profile"

_pharmaceuticals, 2025, doi:10.3390/ph18111617_

Round 1
Reviewer 1 Report
Comments and Suggestions for Authors
This study examines the impact of genetic variations in ADME (absorption, distribution, metabolism, and excretion) genes on the pharmacokinetics and safety profile of DD217, a novel oral anticoagulant. The researchers conducted an analysis of genetic variants in the CYP2C and ABCB1 (P-glycoprotein) genes among patients undergoing knee replacement surgery who were administered DD217. The results indicated that variants in CYP2C9 influence absorption dynamics, while polymorphisms in ABCB1 may modulate systemic exposure and affect thrombotic risk. These findings imply that genetic variations in ADME genes contribute to individual responses to DD217, a hypothesis that warrants further investigation in larger cohorts with extended follow-up periods.
- The study references in silico predictions regarding metabolic pathways and transporter involvement. Could the authors provide a more comprehensive overview of the specific in silico tools and databases utilized, along with the rationale for their selection?
- The article addresses the potential for phenoconversion. Could the authors elaborate on how possible drug-drug interactions might affect the study's outcomes and discuss any strategies employed to mitigate these interactions?
- Although SNP-level associations for ABCB1 were identified, haplotype analysis did not corroborate these findings. Can the authors provide further insight into the reasons for this discrepancy, potentially addressing issues related to linkage disequilibrium or interactions among the SNPs studied?
- The study indicates that CYP2C9 variants influence Tmax. Could the authors discuss the clinical significance of altered Tmax values for DD217, particularly concerning its anticoagulant effects and the timing of thrombotic events?
- The dose-dependent nature of ABCB1 associations is emphasized. Can the authors explore the potential mechanisms that may explain why these effects were observed at a dosage of 60 mg but not at 40 mg, including considerations of saturation kinetics or transporter expression levels?
- Given the limited occurrence of thrombotic and bleeding events, what limitations exist in analyzing potential gene-environment interactions? Would it be advantageous to incorporate additional factors such as age, sex, or concurrent medications in future analyses to enhance statistical power?
- Please provide more detailed inclusion and exclusion criteria for the study participants, particularly concerning co-morbidities and concurrent medication usage.
- Can the authors elaborate on the quality control measures implemented during genotyping to ensure the accuracy and reliability of the data collected?
- The discussion section makes comparisons to other direct oral anticoagulants (DOACs) such as rivaroxaban and apixaban. How does the potential for genetic variability of DD217 compare to these other medications, based on the existing literature?
.10. Is there a biological explanation that the authors can propose to account for the absence of the AG genotype in their samples?
11. The article presents a technical and somewhat dry narrative. It may be beneficial to include a simple figure to visualize key findings, such as a forest plot of significant associations or a network diagram illustrating gene-drug interactions. This could enhance accessibility for a broader audience.
Author Response
Reviewer 1.
Comment 1. The study references in silico predictions regarding metabolic pathways and transporter involvement. Could the authors provide a more comprehensive overview of the specific in silico tools and databases utilized, along with the rationale for their selection?
Response: We have extended this section by including the following text in manuscript:
“…The pharmacological potential of Dimolegin was assessed by predicting its effects, metabolism, and transporter interactions. For pharmacological effect prediction, we used the PASS program [40, 41], a well-established program trained on large, curated structure–activity datasets. PASS was selected because of its broad coverage of pharmacological endpoints, repeated peer‑reviewed validation, and suitability for exploring novel chemotypes (see, e.g.: Abdul-Hammed M. et al. (2022) [42]; Bocharova O.A. et al. (2023) [43]; Gangwal A. et al. (2025) [44]; Medvedeva S.M. et al. (2024) [45]; Muratov E.N. et al. (2020) [46]; Panina S.V. et al. (2022) [47]; Schimunek J. et al (2024) [48]; Sukhachev V.S. et al. (2024) [49]).
…
For pharmacokinetic profiling of DD217 we used various widely issued publicly available online resources, including SwissADME [50], CYPstrate [51], P450 Analyzer [52], CYPlebrity[53], ADMETlab 2.0 [54] and MetaPASS [55]. These tools were chosen based on their predictive performance and their complementarity in terms of metabolic and transporter-related endpoints. The use of multiple prediction tools improves the reliability of the final assessment of the pharmacokinetic profile of DD217….”
We also used ADMETlab 2.0 to predict interactions with P-gp (see, Table 21) and determine whether DD217 is a substrate for cytochrome P450 enzymes.
The relevant references have been added to the revised manuscript.
Comment 2. The article addresses the potential for phenoconversion. Could the authors elaborate on how possible drug-drug interactions might affect the study's outcomes and discuss any strategies employed to mitigate these interactions?
Response: Thank you for raising this important point. In the original clinical trial from which the present study population was derived (ClinicalTrials.gov identifier: NCT05189002, https://clinicaltrials.gov/study/NCT05189002), concomitant administration of drugs known to inhibit or induce CYP3A4, CYP2C9, or P-glycoprotein (P-gp) was explicitly prohibited. According to the trial protocol, subjects receiving systemic anticoagulants, antiplatelet agents, or any CYP3A4/P-gp inhibitors or inducers within 14 days before enrollment were excluded.
Therefore, the influence of drug–drug interactions and potential phenoconversion effects was minimized by design. No participants were exposed to medications capable of altering CYP or transporter activity during the pharmacokinetic assessment period. This ensured that the observed interindividual variability could be primarily attributed to genetic rather than environmental or pharmacological factors.
Comment 3. Although SNP-level associations for ABCB1 were identified, haplotype analysis did not corroborate these findings. Can the authors provide further insight into the reasons for this discrepancy, potentially addressing issues related to linkage disequilibrium or interactions among the SNPs studied?
Response: The haplotype analysis of ABCB1 variants (rs1045642, rs2032582, rs1128503, rs4148738) was performed to ensure comparability with previous pharmacogenetic studies of other direct oral anticoagulants (DOACs), where the so-called “ABCB1 1236–2677–3435 haplotype” has been associated with altered P-glycoprotein (P-gp) activity and drug exposure. Thus, our purpose was to examine whether similar patterns of combined variant effects could be observed for the investigational drug with the same mechanism of action by inhibitiom of Xa and similar molecular structure to the betrixaban anticoagulant.
The observed discrepancy between single-SNP and haplotype analyses likely reflects both genetic and methodological factors. The ABCB1 polymorphisms analyzed (rs1045642, rs2032582, rs1128503, rs4148738) are not in complete linkage disequilibrium (LD) in European populations (r² values 0.3–0.7 according to the 1000 Genomes database), indicating partially independent inheritance patterns. As a result, combining these variants into haplotypes may dilute the individual effects detected at the SNP level.
Moreover, the relatively small sample size limited the statistical power to detect associations for less frequent haplotypes (<10% prevalence). Functionally, the SNPs studied have different biological impacts—rs1045642 affects mRNA stability and P-gp expression, whereas rs2032582 encodes a missense change – suggesting potentially nonadditive or opposing effects when considered jointly.
Therefore, the SNP-level findings likely reflect the contribution of individual functionally relevant variants, while haplotype-based effects may have been masked by LD heterogeneity and limited sample size.
Clarifications regarding the rationale for haplotype analysis and the role of linkage disequilibrium were added to the Discussion section:
«Moreover, only moderate linkage disequilibrium between these loci (r² = 0.3-0.7 in Eu-ropean populations) suggests that their combined evaluation may have diluted the effects observed for individual SNPs.»
Comment 4. The study indicates that CYP2C9 variants influence Tmax. Could the authors discuss the clinical significance of altered Tmax values for DD217, particularly concerning its anticoagulant effects and the timing of thrombotic events?
Response: For direct factor Xa inhibitors, pharmacodynamic (anti-Xa) activity closely parallels plasma concentrations; thus, Tmax approximately corresponds to the time of maximal anticoagulant effect. In our study, carriers of reduced-function CYP2C9 alleles (IM/PM) demonstrated a shorter Tmax (median 1.0 vs. 2.0 h at the 60 mg dose), which may reflect decreased intestinal first-pass metabolism and faster absorption kinetics. Theoretically, an earlier Tmax could result in a more rapid onset of anticoagulation and a narrower “unprotected window” immediately after dosing, which may be clinically relevant in the early postoperative period when the risk of venous thromboembolism is highest.
However, in our cohort no consistent associations were observed between CYP2C9 phenotypes and thrombotic or bleeding events, and the observed differences in AUClast and Cmax did not reach statistical significance. Thus, while the shift in Tmax suggests genotype-related variability in absorption dynamics, its clinical implications under standard dosing remain uncertain.
Clarifications regarding this were added to the Discussion section:
“The observed association between CYP2C9 phenotype and shorter Tmax values may be explained by reduced early-phase clearance and attenuated presystemic metabolism, resulting in more rapid and pronounced plasma concentration increases (Cmax) together with a tendency toward greater overall exposure (AUC). Although this shift could theoretically lead to an earlier onset of anticoagulant action, no consistent relationship with thrombotic or bleeding events was observed in our cohort. Further studies including time-matched anti-Xa pharmacokinetic/pharmacodynamic profiling and larger patient samples are warranted to clarify whether earlier peak exposure translates into clinically meaningful differences in DD217’s anticoagulant efficacy or safety.”
Comment 5. The dose-dependent nature of ABCB1 associations is emphasized. Can the authors explore the potential mechanisms that may explain why these effects were observed at a dosage of 60 mg but not at 40 mg, including considerations of saturation kinetics or transporter expression levels?
Response: We thank the reviewer for this insightful question. The Discussion section of the revised manuscript already addresses this issue. The dose-dependent pattern of ABCB1 associations was interpreted as a reflection of saturation or threshold-level involvement of P-gp at higher plasma concentrations (60 mg), where the transporter becomes a more relevant limiter of absorption and interindividual differences in its activity translate more strongly into variability in systemic exposure. This interpretation is consistent with data for other P-gp substrates such as rivaroxaban and apixaban, for which genotype effects are more pronounced at higher doses or in the presence of partial transporter saturation.
The corresponding paragraph in the Discussion has been retained and slightly edited for clarity.
Comment 6. Given the limited occurrence of thrombotic and bleeding events, what limitations exist in analyzing potential gene-environment interactions? Would it be advantageous to incorporate additional factors such as age, sex, or concurrent medications in future analyses to enhance statistical power?
Response: We fully agree with the reviewer’s observation. The limited number of thromboembolic and bleeding events (7 and 2, respectively) restricted the statistical power to explore gene–environment interactions or to adjust for multiple covariates. The present study was designed as a proof-of-concept pharmacogenetic analysis primarily focused on identifying potential genetic determinants of DD217 pharmacokinetics and safety rather than developing multivariable predictive models.
Nonetheless, we recognize that factors such as age, sex, renal function, and concurrent medications can modulate both drug exposure and clinical outcomes.
This issue is already discussed in the “Limitations” section of the manuscript. This section explicitly acknowledges that the small number of outcomes precluded a reliable assessment of the influence of genetic factors on clinical complications and emphasizes the need for larger studies incorporating multifactorial models integrating both genetic and clinical determinants of risk. Minor edits were made to clarify this point.
Comment 7. Please provide more detailed inclusion and exclusion criteria for the study participants, particularly concerning co-morbidities and concurrent medication usage.
Response: The requested information is already provided in Section 2.1 (Study Population). As noted in the revised manuscript, the current pharmacogenetic analysis was based on biological samples and de-identified data from participants of the original clinical trial “Evaluation of safety and tolerability of Dimolegin for prevention of thrombotic complications after major orthopedic surgery” (ClinicalTrials.gov identifier: NCT05189002, https://clinicaltrials.gov/study/NCT05189002). The key exclusion criteria of that study included severe renal or hepatic impairment, active bleeding or high bleeding risk, malignant disease, and the use of systemic anticoagulants, antiplatelet agents, or CYP3A4/P-gp inhibitors or inducers within 14 days prior to enrollment. Therefore, potential confounding by comorbidities or concomitant medications was minimized by design.
Comment 8. Can the authors elaborate on the quality control measures implemented during genotyping to ensure the accuracy and reliability of the data collected?
Response: The description of quality control procedures for genotyping has been expanded in the Materials and Methods section. The revised text now specifies the use of positive and negative controls for each PCR run, duplicate testing of approximately 10% of samples (100% concordance), verification of DNA purity, ensuring data reliability.
Comment 9. The discussion section makes comparisons to other direct oral anticoagulants (DOACs) such as rivaroxaban and apixaban. How does the potential for genetic variability of DD217 compare to these other medications, based on the existing literature?
Response: In brief, single-SNP effects reported for rivaroxaban and apixaban are generally modest, often involving transporter genes (ABCB1/ABCG2) with inconsistent replication across cohorts, whereas CYP3A variants rarely demonstrate robust associations. Dabigatran remains the DOAC with the most reproducible pharmacogenetic signal through CES1 (e.g., rs2244613) with links to plasma levels and bleeding risk. Betrixaban - biopharmaceutically similar to DD217 and a recognized P-gp substrate - has limited human PGx evidence but supports transporter-driven variability. In this context, DD217 shows a concordant pattern: ABCB1 variants modulate exposure (AUClast/Cmax), while CYP2C9 phenotypes shift absorption dynamics (Tmax). The overall magnitude of these effects appears modest and comparable to that reported for rivaroxaban/apixaban; as with other DOACs, clinical implications remain uncertain pending larger cohorts and PK/PD integration.
To better reflect this context, we have added a short clarification stating that the magnitude of the observed genotype effects for DD217 is comparable to those described for other DOACs and supports the feasibility of early pharmacogenetic evaluation rather than indicating immediate clinical applicability. The concluding paragraph of the Discussion has been revised accordingly.
Comment 10. Is there a biological explanation that the authors can propose to account for the absence of the AG genotype in their samples?
Response: The absence of the AG genotype for CYP3A4*18 (rs28371759) is expected for a predominantly Russian (European-ancestry) population. Although participant ethnicity could not be formally established due to data de-identification, reference population databases (1000 Genomes, gnomAD) indicate that the minor G allele is virtually absent in European populations (allele frequency -> 0%) and occurs almost exclusively in East Asian groups (2–6%).
Comment 11. The article presents a technical and somewhat dry narrative. It may be beneficial to include a simple figure to visualize key findings, such as a forest plot of significant associations or a network diagram illustrating gene-drug interactions. This could enhance accessibility for a broader audience?
Response: We thank the reviewer for this thoughtful suggestion. To enhance the accessibility and visual clarity of the manuscript, we have prepared a Graphical Abstract summarizing the key pharmacogenetic associations observed in the study. This visual highlights the relationships between CYP2C9 and ABCB1 variants and the corresponding pharmacokinetic parameters of DD217. We believe that the addition of this concise visual summary will improve the readability of the paper and assist readers in grasping the main findings at a glance.
Reviewer 2 Report
Comments and Suggestions for Authors
In the present manuscript, the role of ADME agents’ genetic variation on DD217 pharmacokinetics and safety profile. In general, the manuscript is written well, and the authors have performed extensive work to draw conclusions in the manuscript. However, there are major comments that requires author’s attention as described below:
- In general, the involvement of particular enzymes or transporters involvement in the drug disposition is performed through in vitro systems expressing those enzymes and calculating the metabolic activity. The authors have performed genotyping in this specific case. Please explain the rationale.
- Abstract: IM, PM and TT in the abstract were not defined, please ensure to define abbreviations at the first instance of their appearance.
- Abstract: The authors mentioned that CYP2C9 variants influence absorption dynamics (Tmax). Typically, absorption dynamics are associated in the gastrointestinal tract. Hence kindly explain if CYP2C9 is present in the GIT.
- Introduction: The authors have written that DOAC therapy is associated with the interindividual variability in pharmacological response. Was this response significant that it requires dose adjustments? Please explain with examples of other DOAC’s that are currently in the market. What was pharmacokinetic variability associated with DD217 from the clinical studies?
- Table 9: In case of CYP2C9, poor metabolizers in 40mg have lower Cmax and AUC as compared to normal and intermediate metabolizers. Please explain as such observation was not seen in case of 60mg.
- Table 9: In order to draw specific conclusions based on the data presented, the number of subjects in each category varying significantly. Kindly explain the impact of less and variable number of subjects in each category on interpretability of these results. It is understandable that the data has limited population. However, kindly assess and explain impact of the same in the discussion section.
- Table 17: While it is a good idea to correlate metabolic variants with the adverse events, again the number of adverse events is very less (in few cases it is 0 or 1 or maximum of 3). Hence kindly explain the interpretability of the same.
- Did the authors perform impact of male/female volunteers on the PK parameters or related phenotyping? Kindly explain in the discussion section.
- Kindly explain the impact of this work in the real-world scenario. Are the authors suggesting for dose adjustments based on phenotyping? If the product specific guidance to be published, does the authors suggest that specific screening is required for volunteers?
Author Response
Comment 1. In general, the involvement of particular enzymes or transporters involvement in the drug disposition is performed through in vitro systems expressing those enzymes and calculating the metabolic activity. The authors have performed genotyping in this specific case. Please explain the rationale.
Response: We thank the reviewer for this insightful comment. The present work was conceived as a pilot, hypothesis-generating study rather than a mechanistic investigation, representing a secondary pharmacogenetic analysis based on biological samples and clinical data from a completed phase II study (ClinicalTrials.gov identifier: NCT05189002, https://clinicaltrials.gov/study/NCT05189002). Therefore, in vitro experiments were beyond the scope of this investigation. Our aim was not to quantify enzymatic activity directly but to explore whether in silico-predicted metabolic and transporter pathways for DD217 show measurable pharmacogenetic signals in a clinical setting.
We consider it scientifically valuable that pharmacogenetic evaluation was initiated at such an early stage in the drug’s life cycle, as most pharmacogenetic studies of DOACs have historically been performed only after extensive clinical use. This approach follows FDA and EMA recommendations encouraging the integration of pharmacogenomic assessments into early-stage clinical development to identify potential variability drivers before large-scale trials. Genotyping thus served as a pragmatic and translational tool to test the biological plausibility of in silico predictions in humans, helping prioritize specific ADME targets for subsequent mechanistic validation in dedicated in vitro systems.
We believe early incorporation of such approaches may help identify sources of pharmacokinetic variability sooner and stimulate further research in this field.
Comment 2. Abstract: IM, PM and TT in the abstract were not defined, please ensure to define abbreviations at the first instance of their appearance.
Response: The abbreviations have been defined at their first mention in the Abstract as follows: IM – intermediate metabolizers, PM – poor metabolizers, and TT – wild-type homozygotes.
Comment 3. Abstract: The authors mentioned that CYP2C9 variants influence absorption dynamics (Tmax). Typically, absorption dynamics are associated in the gastrointestinal tract. Hence kindly explain if CYP2C9 is present in the GIT.
Response: We thank the reviewer for this valuable observation. We agree that the original wording in the Abstract was inaccurate and could have been misinterpreted as implying a direct role of CYP2C9 in intestinal absorption. Our intention was to indicate that CYP2C9 variants were associated with differences in the early phase of the pharmacokinetic profile (Tmax), rather than with the absorption process itself. The Conclusions in Abstract have been revised accordingly:
“Conclusions: This first pharmacogenetic evaluation of DD217 shows that CYP2C9 variants are associated with differences in early-phase pharmacokinetics (Tmax), while ABCB1 polymorphisms appear to modulate systemic exposure (AUClast, Cmax) and may in-fluence thrombotic risk. These observations are consistent with in silico predictions of metabolic and transporter pathways. Despite limitations in sample size and event fre-quency, the study highlights the feasibility and importance of early pharmacogenetic evaluation during the drug development cycle of novel DOACs.”
Comment 4. Introduction: The authors have written that DOAC therapy is associated with the interindividual variability in pharmacological response. Was this response significant that it requires dose adjustments? Please explain with examples of other DOAC’s that are currently in the market. What was pharmacokinetic variability associated with DD217 from the clinical studies?
Response: Interindividual variability is a recognized feature across marketed DOACs; however, routine dose adjustments are guided primarily by clinical covariates (e.g., renal function, age, body weight, and drug–drug interactions with strong P-gp/CYP3A inhibitors/inducers), rather than by germline pharmacogenetics. For example, apixaban is reduced to 2.5 mg twice daily when patients meet two of the following: age ≥80 years, body weight ≤60 kg, or serum creatinine ≥1.5 mg/dL; rivaroxaban and dabigatran have labeled adjustments based mainly on renal function and interacting medications. These label-directed adjustments reflect clinically meaningful variability, yet they are driven by clinical factors, while single-SNP effects have shown modest and inconsistent influence on exposure/outcomes in the literature.
Regarding DD217, in the phase II program (fixed doses 40 and 60 mg once daily) between-subject PK variability was evident. In our pharmacogenetic subset, we observed genotype-associated differences predominantly in Tmax for CYP2C9 and exposure (AUClast, Cmax) for ABCB1; however, the study was not designed for dose-finding or label-type adjustment criteria, and the sample size precludes definitive conclusions about clinical dosing implications - not to propose dose adjustment strategies.
Comment 5. Table 9: In case of CYP2C9, poor metabolizers in 40mg have lower Cmax and AUC as compared to normal and intermediate metabolizers. Please explain as such observation was not seen in case of 60mg.
Response: We thank the reviewer for this observation. The apparent difference between the 40 mg and 60 mg dose groups likely reflects a combination of dose-dependent pharmacokinetic effects and limited subgroup size. At the lower dose, variability in absorption and random distribution of clinical covariates (age, renal function, body weight) among metabolizer groups may have contributed to the numerically lower Cmax and AUC in poor metabolizers. In contrast, at 60 mg, partial saturation of CYP2C9-mediated metabolism may have attenuated or reversed this pattern. Given the small number of poor metabolizers in each subgroup, these findings should be regarded as exploratory and not indicative of a consistent dose-phenotype interaction.
To clarify this, we have expanded the corresponding section of the Discussion:
“For CYP2C9, an association between metabolic phenotype and Tmax was observed at the 60 mg/day dose: carriers of reduced-function alleles (IM+PM) exhibited a significantly shorter time to reach Cmax compared with normal metabolizers (NM), alongside a trend toward higher AUC and Cmax values. At the 40 mg dose, no consistent pattern was found, which may reflect a combination of dose-dependent pharmacokinetic effects and limited subgroup size, whereas at 60 mg partial saturation of CYP2C9-mediated clearance could enhance differences in exposure.”
Comment 6. Table 9: In order to draw specific conclusions based on the data presented, the number of subjects in each category varying significantly. Kindly explain the impact of less and variable number of subjects in each category on interpretability of these results. It is understandable that the data has limited population. However, kindly assess and explain impact of the same in the discussion section.
Response: We thank reviewer for this comment. The impact of the limited and uneven distribution of subjects across metabolizer categories was acknowledged in the Discussion and Limitations sections. As noted, the small number of poor metabolizers (PM) reduces statistical power and increases variability in group means, which limits the interpretability of subgroup comparisons.
Because the PM subgroup was small, for statistical purposes we combined intermediate and poor metabolizers (IM + PM) into one group and compared it with normal metabolizers (NM), as shown in the Table 9 (Table 3 according new numeration in revised text). This grouping approach is commonly applied in pharmacogenetic studies to improve robustness when rare phenotypes are underrepresented. Accordingly, the results for CYP2C9 were interpreted as exploratory and should be viewed as hypothesis-generating rather than confirmatory.
Comment 7. Table 17: While it is a good idea to correlate metabolic variants with the adverse events, again the number of adverse events is very less (in few cases it is 0 or 1 or maximum of 3). Hence kindly explain the interpretability of the same.
Response: We agree that the very small number of adverse events limits the interpretability of genotype–outcome correlations. As stated in the Limitations section, only seven thrombotic events and two bleeding episodes were recorded in the study cohort, which precludes any statistically reliable assessment of associations with pharmacogenetic markers.
Therefore, the data presented in Table 17 (Table 11 according new numeration in revised text) are rather descriptive and exploratory, intended to illustrate the analytical approach rather than to support definitive conclusions. These results should be interpreted with caution until validated in larger patient populations with sufficient event rates.
Comment 8. Did the authors perform impact of male/female volunteers on the PK parameters or related phenotyping? Kindly explain in the discussion section.
Response: We performed an exploratory comparison of pharmacokinetic parameters (AUC, Cmax, Tmax) between male and female participants for both DD217 dose groups (40 mg and 60 mg). The analysis showed no statistically significant sex-related differences in any parameter (all p > 0.05).
|
DD217 Dose |
Tmax |
p-value |
AUClast |
p-value |
Cmax |
p-value |
|||
|
Man (n=5) |
Woman (n=29) |
Man (n=5) |
Woman (n=29) |
Man (n=5) |
Woman (n=29) |
||||
|
40 mg |
12.0 |
10.0 |
0.783 |
21.74 |
47.02 |
0.611 |
2.01 |
6.21 |
0.439 |
|
60 mg |
24.0 |
7.19 |
0.055 |
14.07 |
74.25 |
0.157 |
0.88 |
6.71 |
0.052 |
A slight trend toward higher Cmax and AUC values and shorter Tmax in females at the 60 mg dose did not reach statistical significance (p = 0.052 and 0,055, respectively). However, the sex distribution in the study was markedly unbalanced - only five male participants (two in the 60 mg group and three in the 40 mg group) compared with 29 females (16 and 13, respectively). Therefore, the statistical power of this comparison was limited.
To enhance transparency, individual-level demographic data (age, sex, BMI) have been added to Table 17 alongside PK parameters, allowing independent verification of these results.
Comment 9. Kindly explain the impact of this work in the real-world scenario. Are the authors suggesting for dose adjustments based on phenotyping? If the product specific guidance to be published, does the authors suggest that specific screening is required for volunteers?
Response: This study does not aim to propose pharmacogenetically guided dose adjustments. It represents a proof-of-concept investigation, illustrating that pharmacogenetic analysis can be incorporated early in the development of a novel oral anticoagulant to identify potential sources of pharmacokinetic variability.
At present, no DOAC has approved dosing recommendations based on genetic markers, and clinical decisions remain driven by demographic and clinical covariates such as renal function, age, and concomitant therapy. The current findings are therefore exploratory and not intended for clinical implementation.
In a real-world context, this approach aligns with emerging perspectives on the early incorporation of pharmacogenomic data into drug development, as emphasized by Cross, Turner, Zhang, and Pirmohamed (Pharmacogenomics Journal, 2024. URL: https://www.nature.com/articles/s41397-024-00329-y). These authors highlight that, although genotype-guided dosing is currently established only for warfarin, extending similar principles to DOACs will require robust evidence from translational and prospective studies. Thus, our findings provide a preliminary scientific rationale for such future research but do not support routine genotyping or dose modification at this stage.
Round 2
Reviewer 2 Report
Comments and Suggestions for Authors
I would like to thank the authors for addressing all comments appropriately. I only have one minor comment:
The authors provided explanation for CYP2C9 variants on early phase pharmacokinetics (Tmax). However, this aspect requires further explanation. Is CYP2C9 present in the gut (if yes) is contributing to this behavior? Or is the liver CYP2C9? If Tmax is getting impacted, does it mean Cmax may get impacted as well? Please clarify in the revised manuscript.
Author Response
Comment 1 (Round 2). The authors provided explanation for CYP2C9 variants on early phase pharmacokinetics (Tmax). However, this aspect requires further explanation. Is CYP2C9 present in the gut (if yes) is contributing to this behavior? Or is the liver CYP2C9? If Tmax is getting impacted, does it mean Cmax may get impacted as well? Please clarify in the revised manuscript.
Response: Many thanks to the reviewer for this interesting follow-up comment. CYP2C9 is expressed predominantly in the liver, with only minimal presence in the intestinal mucosa compared for instance with CYP3A4 (for example, Paine MF, Hart HL, Ludington SS, Haining RL, Rettie AE, Zeldin DC. The human intestinal cytochrome P450 "pie". Drug Metab Dispos. 2006;34(5):880-886. doi:10.1124/dmd.105.008672). The observed association between reduced-function CYP2C9 phenotypes (IM/PM) and shorter Tmax at the 60 mg dose most likely reflects a hepatic first-pass effect, rather than intestinal metabolism or altered absorption. In carriers of reduced-activity alleles, lower early hepatic clearance could allow plasma concentrations to rise more rapidly, leading to an earlier Tmax.
Regarding Cmax, although a shift in Tmax might be expected to correspond to higher peaks, the effect did not reach statistical significance in our study cohort (p=0.1259).
Several factors may account for this:
(i) the modest overall contribution of CYP2C9 to DD217 metabolism, as suggested by in silico predictions, implying that genotype-related metabolic differences are relatively small;
(ii) substantial interindividual variability in absorption rate and P-gp-mediated efflux, which predominantly determines the extent of absorption and may obscure small metabolic effects; and
(iii) limited temporal resolution of PK sampling near the concentration peak, which can reduce sensitivity to detect small Cmax differences. Sparse/ inadequate sampling around the peak can bias both Tmax assignment and peak capture, reducing power to detect Cmax differences even when the curve rises earlier. As we know, Tmax is a discrete value (determined by blood sampling points); if samples were taken, for example, every 2 hours, but the real peak occurred after 1 hour and 40 minutes, it will not be recorded, then the peak may be "blurred". This makes it less likely to detect a difference in Cmax.
Taken together, these findings support a biologically plausible but quantitatively minor hepatic contribution of CYP2C9 to early-phase pharmacokinetics, while the lack of a clear Cmax effect might be masked by variability in absorption-related processes.
We believe the current version of the Discussion already reflects this interpretation; therefore, no additional text changes were made.